# The Role of Health Preconditions on COVID-19 Deaths in Portugal: Evidence from Surveillance Data of the First 20293 Infection Cases

**DOI:** 10.3390/jcm9082368

**Published:** 2020-07-24

**Authors:** Paulo Jorge Nogueira, Miguel de Araújo Nobre, Andreia Costa, Ruy M. Ribeiro, Cristina Furtado, Leonor Bacelar Nicolau, Catarina Camarinha, Márcia Luís, Ricardo Abrantes, António Vaz Carneiro

**Affiliations:** 1IMPSP—Instituto de Medicina Preventiva e Saúde Pública, Faculdade de Medicina, Universidade de Lisboa, Avenida Professor Egas Moniz, 1649-028 Lisboa, Portugal; cristina.furtado@insa.min-saude.pt (C.F.); lnicolau@medicina.ulisboa.pt (L.B.N.); avc@medicina.ulisboa.pt (A.V.C.); 2Laboratório de Biomatemática, Faculdade de Medicina, Universidade de Lisboa, Avenida Professor Egas Moniz, 1649-028 Lisboa, Portugal; ruyribeiro@medicina.ulisboa.pt; 3ISBE—Instituto de Saúde Baseada na Evidência, Faculdade de Medicina, Universidade de Lisboa, Avenida Professor Egas Moniz, 1649-028 Lisboa, Portugal; 4ISAMB—Instituto de Saúde Ambiental, Faculdade de Medicina, Universidade de Lisboa, Avenida Professor Egas Moniz, 1649-028 Lisboa, Portugal; andreiajsilvadacosta@gmail.com; 5Clínica Universitária de Estomatologia, Faculdade de Medicina, Universidade de Lisboa, Avenida Professor Egas Moniz, 1649-028 Lisboa, Portugal; mnobre@maloclinics.com; 6UEPID—Unidade de Epidemiologia, Instituto de Medicina Preventiva e Saúde Pública, Faculdade de Medicina, Universidade de Lisboa, Avenida Professor Egas Moniz, 1649-028 Lisboa, Portugal; ccamarinha@medicina.ulisboa.pt (C.C.); marcialuis@campus.ul.pt (M.L.); ricardoabrantes@campus.ul.pt (R.A.); 7ESEL—Escola Superior de Enfermagem de Lisboa, Polo Calouste Gulbenkian Avenida Prof Egas Moniz, 1600-190 Lisboa, Portugal; 8CRC-W—Católica Research Centre for Psychological, Family and Social Wellbeing, Universidade Católica Portuguesa, Palma de Cima, 1649-023 Lisboa, Portugal; 9National Institute of Health Dr. Ricardo Jorge, Av. Padre Cruz, 1600-560 Lisboa, Portugal; 10Cochrane Portugal, Faculdade de Medicina, Universidade de Lisboa, Avenida Professor Egas Moniz, 1649-028 Lisboa, Portugal

**Keywords:** COVID-19, mortality, demographics, systemic condition, cardiovascular, cancer, diabetes, epidemiology, logistic regression, area under the curve

## Abstract

Background: It is essential to study the effect of potential co-factors on the risk of death in patients infected by COVID-19. The identification of risk factors is important to allow more efficient public health and health services strategic interventions with a significant impact on deaths by COVID-19. This study aimed to identify factors associated with COVID-19 deaths in Portugal. Methods: A national dataset with the first 20,293 patients infected with COVID-19 between 1 January and 21 April 2020 was analyzed. The primary outcome measure was mortality by COVID-19, measured (registered and confirmed) by Medical Doctors serving as health delegates on the daily death registry. A logistic regression model using a generalized linear model was used for estimating Odds Ratio (OR) with 95% confidence intervals (95% CI) for each potential risk indicator. Results: A total of 502 infected patients died of COVID-19. The risk factors for increased odds of death by COVID-19 were: sex (male: OR = 1.47, ref = female), age ((56–60) years, OR = 6.01; (61–65) years, OR = 10.5; (66–70) years, OR = 20.4; (71–75) years, OR = 34; (76–80) years, OR = 50.9; (81–85) years, OR = 70.7; (86–90) years, OR = 83.2; (91–95) years, OR = 91.8; (96–104) years, OR = 140.2, ref = (0–55)), Cardiac disease (OR = 2.86), Kidney disorder (OR = 2.95), and Neuromuscular disorder (OR = 1.58), while condition (None (absence of precondition); OR = 0.49) was associated with a reduced chance of dying after adjusting for other variables of interest. Conclusions: Besides age and sex, preconditions justify the risk difference in mortality by COVID-19.

## 1. Introduction

Since December 2019, a pneumonia caused by a novel coronavirus (SARS-CoV-2) has emerged around the globe, leading to important health concerns worldwide [1].

The most prevalent comorbidities in the Corona Virus Disease 2019 (COVID-19) patients have been reported to be hypertension, diabetes, cardiovascular diseases, and respiratory system disorders [2]. Between 11 December 2019 and 31 January 2020, of the 1590 cases analyzed by Guan et al. [3] 25.1% of laboratory-confirmed individuals had at least one comorbidity, with HR of 1.79 (95% CI 1.16–2.77) among patients with at least one comorbidity and 2.59 (95% CI 1.61–4.17) among patients with two or more comorbidities compared with patients without comorbidity. From the 3912 deaths that occurred in England and Wales relating to COVID-19 (March 2020), 3563 (91%) had at least one pre-existing condition [4].

Outcomes such as admission to an intensive care unit (ICU), invasive ventilation, death and greater disease severity have been studied in the context of COVID-19 disease [5]. The findings of these studies suggested that the majority of patients who had moderate to severe respiratory failure required invasive mechanical ventilation and patients with severe illness required admission to the ICU. Furthermore, patients with comorbidity are at higher risks for poor outcomes: respiratory failure, cardiovascular diseases, diabetes, and kidney injury seem to be highly associated with the death of patients with COVID-19 [3,6,7]. Adverse outcomes were also found in older individuals (≥60 years) [4,7,8] and in males [9] in data from both China and European countries. It is important to consider that most studies have been performed in China, which may have different demographic characteristics and prevalence of comorbidities compared to the European population, and therefore a different effect and impact of risk factors [9]. Data describing underlying health conditions among Portuguese patients with COVID-19 have not yet been reported. Limited information has been accessible to describe the presence of comorbidities and outcomes in these individuals.

Globally, there has been an effort to implement several control measures to prevent, prepare, and develop integrated policies against COVID-19. The knowledge of the impact of comorbidities on possible outcomes will allow a more efficient clinical triage of patients by identifying those at greater risk of progression of COVID-19 and more severe complications including death. Furthermore, health services can predict more accurately the need for treatments, facilities, equipment, and necessary staff, according to the risk of patients with a positive test. Considering no vaccine or antiviral treatment for COVID-19 proved effective, public health authorities should prioritize measures and activate specific mechanisms for high-risk patients and strategic community-based networks to protect vulnerable groups of people before and after diagnostic of COVID-19.

In the present study, we performed a comprehensive evaluation of a database with 20,293 COVID-19-infected individuals. The aim of this study was to identify factors and the role of preconditions associated with COVID-19 deaths in Portugal from 1 January to 21 April 2020.

## 2. Materials and Methods

### 2.1. Study Design

This study consisted of the analysis of an official public dataset provided by the General Health Directorate of Portugal (DGS) on 27 April 2020. The dataset is available from the General-Directorate of Health in Portugal (DGS), by request and submission of a detailed analysis project and an ethical committee authorization to perform research on the dataset [10]. Provision of the dataset is then at the discretion of the General Directorate of Health. The data were collected by the SINAVE (National System for Epidemiological Surveillance), which is the national database for mandatory diseases and public health problems notifications, managed by DGS, that provided the data fully anonymized. The research team filled the publicly available form and submitted an analysis project. The project was approved by an Ethics Committee (number 165/2020).

### 2.2. Data Collected

The provided data concern information between 1 January and 21 April 2020 on infected cases and deaths by COVID-19. A confirmed case was based on a positive polymerase chain reaction test. The primary outcome measure was mortality by COVID-19, measured (registered and confirmed) by Medical Doctors serving as health delegates on the daily death registry. The dataset received on 27 April, encompasses confirmed deaths up to 21 April. Potential deaths from cases registered later in the database (in particular between early April and 27 April) are not confirmed and accounted for in the dataset.

As referenced above, the data were registered in SINAVE, an electronic system for epidemiological surveillance of mandatory diseases that includes laboratory notification. SARS-CoV-2 cases are registered by doctors in SINAVE allowing the electronic communication with local, regional, and national health authorities [11]. The SARS-CoV-2 case definition used is based on the World Health Organization case definition.

The SINAVE allows epidemiological surveillance and epidemiological data registration including laboratory notification, ongoing treatment, hospitalization, existing health preconditions (morbidity), and death registration.

Individualized health precondition indicator variables were constructed from the two variables provided in the database (“precondition” and “other precondition”).

### 2.3. General Characteristics and Outcome

The data retrieved include individuals’ demographic characteristics (age, sex, region), COVID-19 disease information (death, recovery, still in treatment, hospitalization, intensive care, respiratory support), and preconditions (Asthma, Cancer, Cardiac disease, Hematological disorder, Diabetes, HIV and other immune deficiency, Kidney disorder, Liver disorder, Neuromuscular disorder, Other precondition and None (absence of precondition)). The data on intensive care and respiratory support include a number of individuals with “unknown” status that we considered to not have had these interventions if they were never hospitalized. This work considers as primary outcome mortality among those with a positive molecular test for COVID-19 on record.

### 2.4. Statistical Analysis

Descriptive statistics, such as absolute and relative frequencies, mean, standard deviations, and medians were used to summarize univariate characteristics.

Bivariable analyses were performed between the outcome variable (death) and variables potentially associated using the X^2^ test or the Fisher exact test. Further bivariate analysis was performed to evaluate the difference in distribution between preconditions and sex, hospitalization, and intensive care.

We analyzed regional differences based on the second level of the Nomenclature of Territorial Units for Statistical purposes (NUTS II) classification for Portugal.

Logistic regression models were performed using a generalized linear model with binomial error distribution and logit link function to estimate the crude and adjusted Odds Ratio (OR) with 95% confidence intervals (95% CI) for each potential risk factor. The model performance was assessed using the area under the curve (AUC) and corresponding 95% confidence intervals (95% CI). The significance level was set at 5%. Statistical analysis was performed using R software version 3.6.3 (R Foundation for Statistical Computing, Vienna, Austria).

## 3. Results

### 3.1. Characteristics of Infected Individuals

The available information comprises all 20,293 individuals reported as infected with SARS-CoV-2 in Portugal between 1 January and 21 April 2020. There were 11,903 female (58.7%) and 8390 male cases (41.3%). The average age (standard deviation) of the individuals was 52.1 years (21.3 years), with a majority between 18 and 55 years of age (57.2%, minimum: 0 years; maximum: 104 years). The majority of cases (60%) occurred in the North region (Table 1). Considering the geographical region, the distribution of infection rates per 100,000 inhabitants was the following: 341.8 in the North region (12,211 individuals infected in a population of 3,572,583); 107.6 in the Algarve region (472 individuals infected in a population of 438,864); 127.1 in the Center region (2817 individuals infected in a population of 2,216,569); 149.8 in the Lisbon Metropolitan region (4264 individuals infected in a population of 2,846,332); 55.4 in the Alentejo region (391 individuals infected in a population of 705,478); 18.9 in the Madeira region (48 individuals infected in a population of 253,945); and 37.1 in the Azores region (90 individuals infected in a population of 242,846).

In terms of health preconditions, the most common observed condition was Diabetes (5.1%), followed by Neuromuscular (3.4%) and Lung disorders (3.4%).

### 3.2. Mortality Among the Infected

There are 502 deaths registered in the database representing an overall lethality of 2.5% of all infected cases at that point (note that cases registered later in the database may not have their final outcome yet). Among the deceased, the mean age was 81.5 ± 10.5 and median 84 years, while in the group that recovered or was still in treatment, mean age was 51.3 ± 21.0 years.

From the available information, at least 2973 (14.7%) infected cases were hospitalized (the data include 1623 cases with unknown hospitalization status), at least 261 (1.3%) infected cases were subjected to intensive care, and at least 86 (0.4%) had respiratory support (oxygen or ventilator). Mortality within the hospitalized cases was 11.1% (330), within those subjected to intensive care 10.3% (27). At this point, no deaths were registered among those submitted to respiratory support (Table 1).

Lethality was higher in men (2.97%) than in women (2.13%). Lethality increased with age reaching 9.5% in infected cases aged 75 to 85 and 12.61% in infected cases older than 85 years old.

Considering the geographical region, the highest lethality was observed in the Centre region with 3.44% (97 deaths), followed by the North region with 2.61% (319 deaths) and Lisbon Metropolitan Area with 1.74% (74 deaths).

### 3.3. Association Between Characteristics and Lethality

Most of the preconditions were significantly associated, in bivariate analyses, with the death outcome (lethality), with increased lethality in those carrying the precondition. The highest lethality was observed among those infected with a prior history of Cardiac and Kidney disorders (Table 1).

### 3.4. Other Associations with Health Precondition

The results of the analyses between preconditions and sex, hospitalization, and intensive care are depicted in Table 2. Most preconditions were significantly more prevalent in men (except for Asthma and None). Hospitalization of infected cases was associated, albeit weakly, with all preconditions. Cardiac disease, Kidney, Neuromuscular, and Hematological disorders had higher chances of hospitalization. Most preconditions were associated with the use of intensive care, with Cardiac disease, Kidney disease, HIV/other immune deficiency, Lung disease, and Diabetes with higher chances of being submitted to intensive care.

### 3.5. Factors Associated with Lethality

The univariable and multivariable logistic regression analysis for the outcome “death” in infected patients with COVID-19 is shown in Table 3 and Figure 1. Table 3 registers three multivariable models: Model 1, a model only including patients without preconditions; Model 2, for each individual precondition adjusted for age and sex; Model 3, the full model. Considering the full multivariable model, males exhibited a 47% increase in mortality compared to female patients. Concerning age, an overall “J” shape along age was defined by the given parameters (odds ratios). After adjusting for the remaining variables, a less defined “J” shape results. This still means that lethality risk increases with the infected cases’ age after adjustment. Considering the preconditions, the majority were bivariately associated with mortality. After adjustment, three preconditions (Cardiac disease, Kidney disease, Neuromuscular disorder, and None (absence of precondition)) remained significantly associated. The presence of Cardiac disease and Kidney disease in the infected cases doubled the chance of mortality by COVID-19. The absence of preconditions (None) showed a protective effect, reducing the chance of mortality by COVID-19. Complementary calculations (multivariable models) were performed with the objective of evaluating the data robustness and model consistency (Table 4). First, we analyzed a model excluding data from the last 3 weeks so that cases with insufficient follow-up were excluded. Next, we analyzed a model including only the Hospitalized cases, assuming that these cases would have more accurate recording of comorbidities. As shown in Table 4, the results of these two reduced models are consistent with the model for the full dataset in Table 3.

The full model performance for predicting COVID-19 mortality is illustrated in Figure 2, where an excellent discrimination capacity was observed (AUC (95% CI): 0.909 (0.900;0.919)).

## 4. Discussion

Our study evaluating a national database of more than 20,000 infected individuals constitutes one of the largest population studies on COVID-19 to date. A total of 14.7% of patients needed hospitalization, associated with a case fatality rate (CFR) of 11.1%, while 1.3% were admitted to the intensive care unit (ICU) with an associated CFR of 10.3%. These figures were somewhat lower when compared to previous systematic reviews, where a 20.3–29.3% and 6.8–13.7% of ICU admission and CFR were reported, respectively [12,13]. This difference may be related with a different stage of the epidemic in Portugal at the time of data analysis, and differences in the timing of non-pharmaceutical actions (schools and medical faculties closures and lockdown declared one and two weeks after the first COVID-19 case, respectively) [14], as the majority of studies included in the systematic reviews were from China. Moreover, it is important to consider the variations of proportions that were found in other individual studies.

This study represents one of the first attempts to understand the lethality of COVID-19 in Portugal from infected cases, proposing risk factors based on multivariable analysis. The risk factors for lethality by COVID-19 were sex (male), advanced age, Kidney disorder, Cardiac disease and Neuromuscular disorder, while the absence of a precondition was associated with a reduced chance of mortality after adjusting for other variables of interest. The principal risk factors for lethality by COVID-19 reported in the present study are supported by the literature. The majority of studies report male patients, older patients, and patients with preconditions at an increased risk of infection and mortality irrespective of the region of the globe [12,15]. A recent meta-analysis of thirteen studies registered male individuals (OR = 1.76), age over 65 years old (OR = 6.06), smoking habits (OR = 2.51), and preconditions including Cardiovascular disease (OR = 5.19), Diabetes (OR = 3.68), Respiratory disease (OR = 3.68), and hypertension (OR = 2.72) as significantly higher in critical/mortal patients compared to non-critical patients [15]. Men registered a 48% increased risk of death by COVID-19 in our study, a result previously observed [9]. We registered an increased prevalence of preconditions in men when compared to women (all preconditions except Asthma); while the absence of preconditions was 30% more likely in female individuals. Moreover, the fact that women could have stronger responses than men in many infectious pathogens [16], the likelihood of women to search for health care services more than men [17], and in the adoption of hygiene practices [18] may further complement the explanation of this result.

Age was an important predictor for mortality in our study, with lethality-adjusted ORs increasing after 55 years of age. Moreover, there was a thirty-year difference in the average age between fatal (average age of 81 years) and non-fatal cases (average age of 51 years). Age was a predictor for mortality in patients both with and without preconditions, a result supported by a previous multicenter cohort study of 191 individuals where increased odds of in-hospital death were associated with older age (odds ratio 1.10, 95% CI 1.03–1.17, per year increase [19]). These results are partially explained by the increased burden of pre-conditions in older age groups, where dramatic increases for the prevalence of preconditions (particularly for Cardiac disease, Kidney disorder, and Neuromuscular disorder) were registered in older age groups when compared to individuals of less than 55 years of age. Nevertheless, age remained a risk factor for death in our study even in patients without preconditions, as would be expected given that advanced age is a risk factor for death even in the absence of COVID-19 infection. Indeed, it is possible that the increase in risk of death is more due to aging than to COVID-19 infection proper. Still, the link between older individuals and the likelihood to develop severe and critical cases of COVID-19 has been made before, either due to immunosenescence, malnutrition, or ignoring more easily the early symptoms and consequently missing the best time to seek medical advice [20,21]. Furthermore, several studies registered an important association between increased age and COVID-19 severity/fatality [22,23,24,25,26]. The vulnerability of elderly individuals is illustrated in recent studies. A retrospective observational study investigating mortality in hospitalized patients with COVID-19 registered the great vulnerability of patients residing in retirement homes, with older age independently associated with mortality when adjusted for other variables of interest [22]. On the opposite residence condition, the shape of COVID-19 vulnerability was estimated based on a random infection of 10% of the population living in private households (excluding individuals living in retirement homes) of 81 countries [27]. In this study, it was estimated that national age and coresidence patterns can alter the vulnerability of a country to COVID-19 outbreaks, with direct effects dependent on a country’s age structure and indirect effects dependent on the size and age structure of a country’s households [27].

The specific comorbidities that emerged as risk factors for mortality in our study (Cardiac disease, Kidney disorder, and Neuromuscular disorder) should be interpreted considering the COVID-19 physiopathology. In our study, the prevalence of Kidney disease on admission in patients with COVID-19 was high and associated with clinical stage decline. The highest in-hospital mortality rate (24.4%) and chances of mortality (2.43-fold increase) were registered for individuals with Kidney disease precondition. Other studies in patients with Kidney disorders recorded comparable results. A meta-analysis of eleven COVID-19 studies registered an association between acute Kidney injury and a higher risk of mortality of almost 16-fold (OR = 15.93), with creatinine levels significantly higher in non-survivors compared to survivors [28]. Nevertheless, the meta-analysis reported a high heterogeneity and a difficulty in adjusting for confounders [28]. Pathophysiological mechanisms may be involved. Considering that CO2 is an independent determinant of pH adjusted by alveolar ventilation, the disturbance of the acid–base regulation (through the interplay of bicarbonate buffer and respiratory and renal systems) may induce acid–base imbalance and in this way may pose a life-threatening situation [29]. A second hypothesis consists in the direct Kidney infection by SARS-CoV-2, which recognizes the human Angiotensin-converting enzyme 2 as a cellular receptor that allows it to infect different host cells, a mechanism previously expressed by SARS-CoV virus [30]. This mechanism could explain the particular importance of acute Kidney injury during hospitalization, considering the exhibited conditions of proteinuria; hematuria; and elevated levels of either serum creatinine, blood urea nitrogen, or both, rendering a significant increase of in-hospital mortality between 1+ and 5.5-fold [31].

Patients with cardiovascular disorder exhibited a nearly 3-fold increase in the chance of dying from COVID-19. Cardiovascular disease has been consistently reported as one of the main risk factors for COVID-19 mortality. Two recent meta-analysis reported an odds ratio (95% CI) of 5.19 (3.25; 8.29) (1) and a risk ratio (95% CI) of 2.25 (1.53; 3.29)) for mortality in cardiovascular disease patients [32]. Our results are in concordance with the systematic review by Pranata et al. [32] and lower than the systematic review of Zheng et al. [15]. A possible reason may be the small study effect on the estimates from Zheng et al. [15], considering the majority of the 13 studies included were of smaller sample size compared to both our study and Pranata et al. [32]. Nevertheless, the direction of the estimate towards risk is clear. The pathophysiological mechanisms behind this association may be multiple: from severe infection with SARS-CoV-2, precipitating myocardial infarction, myocarditis, heart failure, and arrhythmias as well as an acute respiratory distress syndrome and renal failure [33,34]; through the evolution along with multiorgan failure directly due to SARS-CoV-2-infected endothelial cells and resulting endothelitis [33]; to the potential impacts of therapies considering the likely increase in the number of ACE 2 receptors and the corresponding increase in the susceptibility [33]. Furthermore, the link between pneumonia and cardiovascular complications should be accounted for: recent studies explore/registered myocardial injury during SARS-CoV-2, secondary to type 2 myocardial infarction, a consequence of increased oxygen demand or reduced oxygen supply during respiratory failure [34,35,36]. In this scenario, cytokines microvascular activation can cause not only myocardial injury but also harm other systems involved in COVID-19 infections, including the kidneys [37]. Consequently, the relation between cardiovascular disease, Kidney disease and Diabetes should not be ruled out [33,38].

Patients with Neuromuscular disorders registered a 41% increase in lethality. The pathophysiological mechanisms related to this association could be: (i) the fact that patients with this precondition are under the use of immunosuppressive therapies and therefore more likely to increase the severity of COVID-19 infection [39]; and (ii) risk of exacerbation of myasthenia gravis and QT prolongation in patients with pre-existing cardiac involvement secondary to the treatment with hydroxychloroquine and azithromycin [40]. However, given that the present database did not provide the patient-specific pharmacological therapies, this question remains open.

Diabetes, previously registered as a significant risk factor for COVID-19 mortality [3], was not significant in our study when adjusted for other variables of interest. A potential reason for this result might be related to the level of glycemic control. A recent study evaluating the impact of blood glucose control and outcomes of COVID-19 in pre-existing type 2 diabetes noted that when adjusting the model for well-controlled blood glucose, a marked lower mortality was registered compared to individuals with poorly controlled blood glucose [41]. Data from a national study in Portugal with 1688 individuals reported about 82% of Portuguese diabetics were pharmacologically medicated and that 60.7% were controlled [42]. However, since we were not able to retrieve the level of glycemic control in diabetic individuals in our study, this question remains open.

The strengths of this study include being a population-based study, the large sample size, and the origin of the data. SINAVE is the electronic platform for notification and cases monitoring of mandatory communicable diseases, allowing the analysis and evaluation of emergent situations, particularly large-scale epidemic outbreaks and pandemics, such as a COVID-19. This system allows for the electronic articulation of doctors (who notify cases of illness), health authorities (responsible for epidemiology at local, regional and national levels), and laboratories (cases notification and cases confirmation). SINAVE allows real-time notification, admitting the implementation of control measures to control and limit the spread of disease and the occurrence of additional cases. The SINAVE database is relevant for the present COVID-19 study by its quality and extensibility since it is based on the information registered by the Medical Doctor who notified the case. Furthermore, it contains all the notified cases up to the extraction of the data due to the interoperability characteristics between the SINAVE computer application and clinical process computer applications.

The limitations of the present study include the absence of potentially important data from the database, missing data, underreporting of mild cases, the impossibility of accounting for the temporal sequence of events, and under-reporting of preconditions. The limitations are presented and discussed in detail below, including the potential bias and corresponding direction. The database did not include reported symptoms and laboratory test results. The existence of unknown values in the data for some outcomes, together with the reporting of preconditions in the medical record, may lead to an underestimation of some risk indicators. It is likely that some of the preconditions were under-reported both in quantity and importance: a particular example is cerebrovascular disease, considered one of the comorbidities with significant impact in COVID-19 prognostic [7], and conspicuously absent from our database. The temporal sequence of events was not taken into account (time elapsed between the onset of symptoms and hospital admission, or time between hospital admission and death), which may imply an underestimation of preconditions. Finally, because the patients’ clinical observation is still ongoing, many individuals have not reached clinical endpoints (recovery or death). The authors performed complementary calculations to evaluate data robustness and model consistency: The results of these complementary estimations (one model excluding cases from the last 3 weeks and the other model on hospitalized cases) show reasonable consistency with the full multivariable model. A further limitation is related to the data that concern only the initial phase of the pandemic in Portugal up to 21 April. The pandemic is still ongoing, registering on June 25 about 40,866 infected individuals and 1555 deaths [43]. Nevertheless, no update of the data was made available up to the moment, and therefore the hypothesis of a difference in mortality between the initial and actual phase remains open.

The results of the present study registered potential different pathophysiological mechanisms for COVID-19 mortality, suggesting the need for a team approach between different medical specialties in order to maximize the probabilities of recovery for COVID-19 patients. Future research with larger data sets should include the study of effect and impact of preconditions with individuals reaching clinical endpoints to gain a better understanding of risk factors, as well as the economic and health impacts of COVID-19.

## 5. Conclusions

Based on the results, lethality by COVID-19 in Portuguese infected individuals was significantly associated with demographics (males; advanced age) and the preconditions Cardiac disease, Kidney disease, and Neuromuscular disorder. The present study successfully modeled the condition to assess the prognosis of each patient with high precision. Being one of the first studies in Europe not only to identify the main preconditions associated with COVID-19 lethality but also to include a model for individuals with absence of preconditions in more than 20,000 cases, this research thus represents a potential important benchmark for future studies.

## Figures and Tables

**Figure 1 jcm-09-02368-f001:**
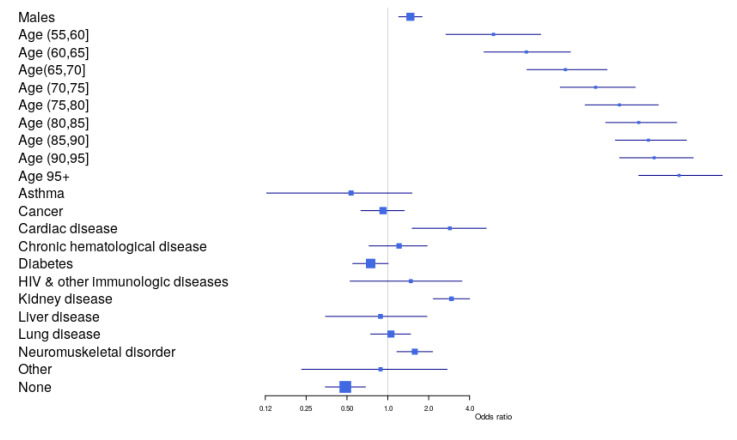
Forest plot illustrating the contribution of each co-factor for the overall model.

**Figure 2 jcm-09-02368-f002:**
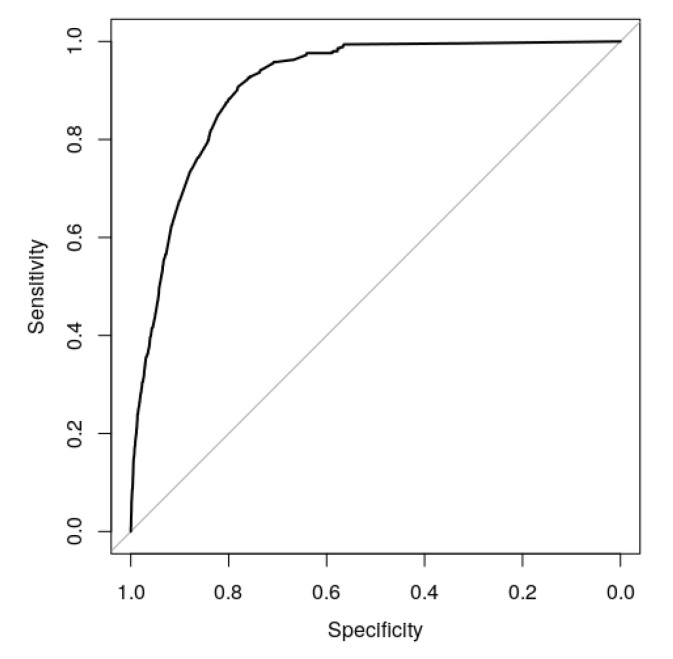
The area under the curve illustrating the performance of the full model to predict mortality by COVID-19. Note the high precision given by the area under the curve (AUC) (95% CI) of 0.909 (0.900;0.919).

**Table 1 jcm-09-02368-t001:** Characteristics of the SARS-CoV-2 infected individuals and corresponding lethality in Portugal on 21 April 2020.

Variable		*n* (%)	Deaths (%)	*p*-Value *
Outcome	Recovered	1244 (6.1%)		
	Died COVID-19	502 (2.5%)		
	Ongoing Treatment	18,524 (91.3%)		
	Unknown	23 (0.1%)		
Sex	Female	11,903 (58.7%)	253 (2.13%)	<0.001
	Male	8390 (41.3%)	249 (2.97%)	
Age	(0,18)	711 (2.5%)	0 (0%)	<0.001
	(19,35)	4153 (20.5%)	0 (0%)	
	(36,45)	3259 (16.1%)	3 (0.09%)	
	(46,55)	3653 (18.1%)	10 (0.27%)	
	(56,65)	3046 (15.1%)	30 (0.98%)	
	(66,75)	1926 (9.5%)	78 (4.05%)	
	(76,85)	1864 (9.2%)	177 (9.50%)	
	86+	1618 (8.0%)	204 (12.61%)	
Region	North	12211 (60.2%)	319 (2.61%)	<0.001
	Algarve	472 (2.3%)	6 (1.27%)	
	Center	2817 (13.9%)	97 (3.44%)	
	Lisbon Metropolitan Area	4264 (21.0%)	74 (1.74%)	
	Alentejo	391 (1.9%)	6 (1.53%)	
	Madeira	48 (0.2%)	0 (0%)	
	Azores	90 (0.4%)	0 (0%)	
Hospitalization	No	15,697 (77.4%)	126 (0.80%)	<0.001
	Unknown	1623 (8.0%)	46 (2.83%)	
	Yes	2973 (14.7%)	330 (11.10%)	
Intensive Care	No	15,697 (77.4%)	126 (0.80%)	<0.001
	Unknown	4335 (21.4%)	475 (10.96%)	
	Yes	261 (1.3%)	27 (10.34%)	
Respiratory Support	No	1315 (6.5%)	156 (11.86%)	<0.001
	Oxygen	59 (0.3%)	0 (0%)	
	Ventilator	26 (0.1%)	0 (0%)	
	Unknown	18,893 (93.1%)	346 (1.83%)	
Precondition				
Asthma	Presence	277 (1.4%)	3 (1.08%)	0.192
	Absence	20,016 (98.7%)	499 (2.50%)	
Cancer	Presence	611 (3.0%)	47 (7.69%)	<0.001
	Absence	19,682 (97.0%)	455 (2.3%)	
Cardiac Disease	Presence	54 (0.3%)	19 (35.2%)	<0.001
	Absence	20,239 (99.7%)	483 (2.4%)	
Hematological Disorder	Presence	220 (1.1%)	29 (13.2%)	<0.001
	Absence	20,073 (98.9%)	473 (2.4%)	
Diabetes	Presence	1145 (5.6%)	83 (7.3%)	<0.001
	Absence	19,148 (94.4%)	419 (2.2%)	
HIV/other Immune Deficiency	Presence	107 (0.5%)	6 (5.6%)	0.075
	Absence	20,186 (99.5%)	496 (2.5%)	
Kidney Disorder	Presence	401 (2.0%)	98 (24.4%)	<0.001
	Absence	19,892 (98.0%)	404 (2.0%)	
Liver Disorder	Presence	107 (0.5%)	7 (6.5%)	0.016
	Absence	20,186 (99.5%)	495 (2.5%)	
Lung Disorder	Presence	688 (3.4%)	60 (8.7%)	<0.001
	Absence	19,605 (96.6%)	442 (2.3%)	
Neuromuscular Disorder	Presence	795 (3.9%)	123 (15.5%)	<0.001
	Absence	19,498 (96.1%)	379 (1.9%)	
Other Condition	Presence	76 (0.4%)	4 (5.3%)	0.231
	Absence	20,217 (99.6%)	498 (2.5%)	
None	No Precondition	16,927 (83.4%)	212 (1.3%)	<0.001
(Absence of Precondition)	At least one Precondition	3366 (16.6%)	290 (8.6%)	

* The *p*-value refers to the comparison between mortality proportions.

**Table 2 jcm-09-02368-t002:** Analysis for sex, hospitalization, and intensive care, according to the individuals’ precondition.

Variables	Sex Odds Ratio (OR) (95% CI) ^1^	*p*-Value	Hospitalization Odds Ratio (OR) (95% CI) ^2^	*p*-Value	Intensive Care Odds Ratio (OR) (95% CI) ^3^	*p*-Value
Asthma	0.71 (0.55;0.92)	*p* = 0.008	0.59 (0.38;0.87)	*p* = 0.011	1.13 (0.30;2.95)	*p* = 0.784
Cancer	1.61 (1.37;1.90)	*p* < 0.001	5.85 (4.94;6.92)	*p* < 0.001	3.04 (1.85;4.75)	*p* < 0.001
Cardiac Disease	1.65 (0.93;2.94)	*p* = 0.072	87.66 (32.19;361)	*p* < 0.001	18.09 (8.02;37.0)	*p* < 0.001
Hematological	1.23 (0.93;1.62)	*p* = 0.130	10.15 (7.61;13.64)	*p* < 0.001	2.96 (1.25;6.02)	*p* = 0.008
Diabetes	1.51 (1.33;1.70)	*p* < 0.001	5.40 (4.75;6.14)	*p* < 0.001	4.42 (3.18;6.04)	*p* < 0.001
HIV/other imune deficiency	1.96 (1.31;2.95)	*p* = 0.009	4.10 (2.74;6.10)	*p* < 0.001	5.49 (2.13;11.9)	*p* < 0.001
Kidney Disorder	2.02 (1.64;2.48)	*p* < 0.001	14.33 (11.48;18.01)	*p* < 0.001	8.02 (5.34;11.74)	*p* < 0.001
Liver Disorder	3.50 (2.27;5.51)	*p* < 0.001	9.07 (6.09;13.68)	*p* < 0.001	3.82 (1.20;9.31)	*p* = 0.012
Lung Disorder	1.51 (1.29;1.76)	*p* < 0.001	4.85 (4.12;5.69)	*p* < 0.001	4.92 (3.34;7.06)	*p* < 0.001
Neuromuscular	1.01 (0.87;1.17)	*p* = 0.883	11.48 (9.82;13.45)	*p* < 0.001	2.65 (1.67;4.04)	*p* < 0.001
Other Condition	1.03 (0.63;1.67)	*p* = 0.907	5.18 (3.24;8.28)	*p* < 0.001	2.08 (0.25;7.87)	*p* = 0.256
None	0.77 (0.71;0.82)	*p* < 0.001	0.13 (0.12;0.14)	*p* < 0.001	0.17 (0.13;0.21)	*p* < 0.001

^1^ Reference- Female; ^2^ Reference- Hospitalization: No; ^3^ Reference- Intensive care: Unknown.

**Table 3 jcm-09-02368-t003:** Univariable and Multivariable logistic regression analyses for the outcome “death” in infected individuals with COVID-19.

Variables	Odds Ratio (OR) Crude Values (95% CI) ^	*p*-Value	Odds Ratio (OR) Adjusted Values (95% CI) ^1^	*p*-Value	Odds Ratio (OR) Adjusted Values (95% CI) ^2^	*p*-Value	Odds Ratio (OR) Adjusted Values (95% CI) ^3^	*p*-Value
Sex								
Female	1.0 (reference)		1.0 (reference)				1.0 (reference)	
Male	1.41 (1.17;1.69)	*p* < 0.001	1.99 (1.49;2.65)	<0.001			1.47 (1.20;1.79)	<0.001
Age (0–55) years	1.0 (reference)		1.0 (reference)	<0.001			1.0 (reference)	
Age (56–60) years	6.50 (2.92;14.36)	*p* < 0.001	13.20 (3.23;64.41)	<0.001			6.01 (2.68;13.40)	<0.001
Age (61–65) years	12.25 (6.02;25.59)	*p* < 0.001	29.57 (8.81;133.44)	<0.001			10.50 (5.12;22.17)	<0.001
Age (66–70) years	26.53 (13.98;53.09)	*p* < 0.001	53.71 (16.72;237.98)	<0.001			20.36 (10.59;41.31)	<0.001
Age (71–75) years	51.22 (28.60;98.67)	*p* < 0.001	115.37 (39.16;492.19)	<0.001			34.01 (18.64;66.69)	<0.001
Age (76–80) years	80.27 (45.75;152.50)	*p* < 0.001	187.76 (65.80;789.55)	<0.001			50.91 (28.46;98.50)	<0.001
Age (81–85) years	108.56 (63.20;203.27)	*p* < 0.001	281.18 (102.16;1162.9)	<0.001			70.65 (40.35;134.69)	<0.001
Age (86–90) years	125.87 (73.28;235.68)	*p* < 0.001	357.87 (130.42;1478.17)	<0.001			83.23 (47.51;58.70)	<0.001
Age (91–95) years	125.67 (71.30;239.55)	*p* < 0.001	407.45 (145.58;1698.79)	<0.001			91.83 (51.10;178.31)	<0.001
Age (96–104) years	183.30 (94.32;374.16)	*p* < 0.001	640.17 (203.95;2818.60)	<0.001			140.17 (70.69;291.53)	<0.001
Asthma	0.03 (0.02;0.03)	*p* = 0.145			0.80 (0.19;2.21)	0.740	0.54 (0.13;1.51)	0.305
Cancer	3.52 (2.55;4.76)	*p* < 0.001			1.48 (1.06;2.04)	0.018	0.92 (0.64;1.32)	0.666
Cardiac Disease	22.20 (12.38;38.65)	*p* < 0.001			6.40 (3.48;11.51)	<0.001	2.86 (1.51;5.32)	<0.001
Chronic Hematological Disorder	6.29 (4.13;9.24)	*p* < 0.001			2.33 (1.50;3.51)	<0.001	1.21 (0.73;1.95)	0.447
Diabetes	3.49 (2.72;4.43)	*p* < 0.001			1.39 (1.08;1.79)	0.010	0.75 (0.55;1.01)	0.057
HIV/Other Imune Deficiency	2.36 (0.92;4.95)	*p* = 0.042			3.12 (1.15;7.19)	0.014	1.48 (0.53;3.52)	0.414
Kidney Disorder	15.60 (12.13;19.93)	*p* < 0.001			4.97 (3.80;6.46)	<0.001	2.95 (2.16;4.00)	<0.001
Liver Disorder	2.79 (1.17;5.60)	*p* = 0.010			1.77 (0.72;3.76)	0.168	0.88 (0.35;1.94)	0.773
Lung Disorder	4.14 (3.10;5.44)	*p* < 0.001			1.79 (1.32;2.39)	<0.001	1.05 (0.75;1.47)	0.761
Neuromuscular Disorder	9.23 (7.41;11.44)	*p* < 0.001			2.67 (2.11;3.34)	<0.001	1.58 (1.17;2.14)	0.003
None	0.14 (0.11;0.16)	*p* < 0.001			0.34 (0.28;0.41)	<0.001	0.49 (0.35;0.68)	<0.001
Other condition	2.20 (0.67;5.33)	*p* = 0.126			1.31 (0.38;3.43)	0.625	0.88 (0.23;2.74)	0.341

^ Univariable analyses; ^1^ Model including only individuals without preconditions; ^2^ Model for each individual precondition adjusted for Age and Sex; ^3^ Full model adjusted for Pregnancy.

**Table 4 jcm-09-02368-t004:** Complementary calculations to evaluate data robustness and model consistency: A multivariable model only including cases up to 31 March 2020 and a multivariable model for hospitalized cases.

Variables	Odds Ratio (OR) Adjusted Values (95% CI) *	*p*-Value	Odds Ratio (OR) Adjusted Values (95% CI) **	*p*-Value
Sex				
Female	1.0 (reference)		1.0 (reference)	
Male	1.59 (1.24;2.06)	<0.001	1.05 (0.81;1.35)	=0.713
Age (0–55) years	1.0 (reference)		1.0 (reference)	
Age (56–60) years	8.29 (3.41;20.77)	<0.001	2.94 (1.13;7.56)	=0.024
Age (61–65) years	12.26 (5.40;29.52)	<0.001	2.53 (1.01;6.37)	0.045
Age (66–70) years	21.76 (10.05;51.01)	<0.001	3.96 (1.78;9.33)	<0.001
Age (71–75) years	37.94 (18.68;85.59)	<0.001	5.62 (2.76;12.53)	<0.001
Age (76–80) years	67.09 (33.77;149.30)	<0.001	8.37 (4.22;18.34)	<0.001
Age (81–85) years	92.48 (47.10;204.31)	<0.001	12.82 (6.65;27.55)	<0.001
Age (86–90) years	141.25 (71.63;313.08)	<0.001	14.22 (7.33;30.72)	<0.001
Age (91–95) years	174.32 (83.94;400.32)	<0.001	14.77 (7.25;33.00)	<0.001
Age (96–104) years	347.63 (140.08;913.73)	<0.001	18.32 (7.37;46.94)	<0.001
Asthma	1.01 (0.23;3.09)	=0.994	0.38 (0.02;2.00)	=0.363
Cancer	1.02 (0.64;1.59)	=0.939	0.98 (0.64;1.48)	=0.930
Cardiac Disease	2.09 (0.99;4.25)	=0.046	2.14 (1.09;4.09)	=0.024
Chronic Hematological Disorder	0.96 (0.49;1.79)	=0.907	1.53 (0.89;2.56)	=0.118
Diabetes	0.63 (0.42;0.94)	=0.027	0.69 (0.49;0.98)	=0.040
HIV/other imune deficiency	2.17 (0.74;5.59)	=0.129	1.27 (0.40;3.32)	=0.650
Kidney Disorder	1.98 (1.35;2.91)	<0.001	2.84 (1.99;4.04)	<0.001
Liver Disorder	1.79 (0.49;5.18)	=0.324	0.88 (0.34;1.98)	=0.779
Lung Disorder	1.04 (0.69;1.57)	=0.844	0.97 (0.64;1.44)	=0.872
Neuromuscular Disorder	1.44 (0.98;2.11)	=0.065	1.39 (0.98;1.97)	=0.066
None	0.59 (0.38;0.90)	=0.014	0.84 (0.55;1.27)	=0.394
Other condition	1.05 (0.14;5.00)	=0.960	0.57 (0.12;2.04)	=0.426

* Full model with cases up to 31 March (*n* = 11103 individuals; n = 328 deaths); ** Model of Hospitalized cases (*n* = 2973 individuals; *n* = 230 deaths).

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
