# Peer review of "The Role of Health Preconditions on COVID-19 Deaths in Portugal: Evidence from Surveillance Data of the First 20293 Infection Cases"

_jcm, 2020, doi:10.3390/jcm9082368_

Round 1
Reviewer 1 Report
This important study reporting more than 20 000 COVID-19 patients from the Portuguese National database is of major interest in the current context.
However, I would like to suggest some revisions, mainly in the form.
Major revision :
1- It is not clear when was mesured the primary outcome : mortality, but at what time ? The authors must clarify this point.
I suppose it is at the time of the data extraction. If so, follow-up time varies between patients at the beginning and at the end of the inclusion period and this should be discussed.
2- I suggest the authors add a flowchart, or at least more clearly indicate the number of patients with lacking data.
Minor revisions :
1- ABSTRACT
From my point of view, the abstract could be improved :
- Briefly explain why the identification of risk factors of death is important.
- Clearly indicate when the primary outcome was mesured.
- "with binomial error distribution and logit link function", "The significant level was set at 5%" : These methods elements are probably not essential in the abstract.
- Rather than indicate the AUC of the model in the abstract, I would suggest the authors indicate the OR for identified risk factors. Indeed, this study aimed to identify risk factors for death, not to create a pronostic model.
2- RESULTS
The tables are difficult to read in the present form .
1) Please supress all the unneccessary points in the tables.
2) In table 1, please clarify or suppress the part "outcome" of the table: What do you mean with "ongoing treatment" for 91.3% of patients ? It is unlikely that patients infected in January are still under specific treatment, though this depends on when the data are extracted.
3) In table 3, please supress the full lines between the different age classes.
Figure 1, please explicit the abbreviations, or better change the name of the variables.
DISCUSSION :
Line 22: age (continuous..age>75 ans) + Line 23 "age (18-55)" > I would simplify by indicating that advanced age is a risk factor.
Line 50: Wether age is per se a risk factor for death in pneumonia after adjustment for comorbidities remain a debated question. I would suggest further discussing it in light of the current literature.
Line 81: The authors clearly highlight that preexisiting cardiovascular disease remained the main risk factor for death. Please further discuss the link between pneumonia and cardiovascular complications, including type 2 myocardial infarction, particularly frequent in this older population.
CONCLUSIONS :
I am not sure the distinction between age over 75y and age 18-55 is relevant.
Please indicate Author Contributions and Funding sources.
Author Response
Response to Review of manuscript no. jcm-849655, “The role of health preconditions on COVID-19 deaths in Portugal: Evidence from surveillance data of the first 20293 infection cases”
Reviewer 1
Major revision:
1- It is not clear when was measured the primary outcome: mortality, but at what time? The authors must clarify this point.
I suppose it is at the time of the data extraction. If so, follow-up time varies between patients at the beginning and at the end of the inclusion period and this should be discussed.
Response: The authors thank the Reviewer’s query. The primary outcome was measured (registered and confirmed) by Medical Doctors serving as health delegates on the daily death registry. The information was not available in the dataset nor the authors received the information from the Health Directorate despite several contacts. Therefore, it was assumed that the primary outcome measure was measured in the period between January 1 and April 9, 2020 as indicated in the manuscript. The authors performed complementary calculations to address this issue, introducing two multivariable models: one model for the cases up to March 31, 2020 (excluding cases from the last 3 weeks) where the effect of age on death by COVID-19 increased; and another model for Hospitalized cases, where both age and precondition effects decreased. The general results of these models point to a reasonable consistency of the full multivariable model irrespective of the dates limitation. The authors introduced the necessary information to make the message clearer to the reader.
Changes: Abstract section, line 27: “A national dataset with the first 20293 patients infected with COVID-19 between January 1 and April 9, 2020 was analyzed. “
Line 28: “The primary outcome was measured (registered and confirmed) by Medical Doctors serving as health delegates on the daily death registry.“
Results section, line 209-214: “Complementary calculations (multivariable models) were performed with the objective of evaluating the data robustness and model consistency (Table 4). First, we analyzed a model excluding data from the last 3 weeks, so that cases with insufficient follow-up were excluded. Next, we analyzed a model including only the Hospitalized cases, assuming that these cases would have more accurate recording of comorbidities. As shown in Table 4, the results of these two reduced models are consistent with the model for the full dataset in Table 3.”
Results section, line 6: Table 4.
Discussion section, line 157-164: “The authors performed complementary calculations to evaluate data robustness and model consistency: The results of these complementary estimations (one model excluding cases from the last 3 weeks and the other model on hospitalized cases) show reasonable consistency of the full multivariable model. A further limitation is related to the data that concerns the initial phase of the pandemic in Portugal up to April 9. The pandemic is still ongoing, registering on June 25 about 40866 infected individuals and 1555 deaths [43]. Nevertheless, no update of the data was made available up to the moment and therefore the hypothesis of a difference in mortality between the initial and actual phase remains open.”
2- I suggest the authors add a flowchart, or at least more clearly indicate the number of patients with lacking data.
Response: The authors thank the Reviewer’s suggestion. A flowchart would be more useful if this research team was the one to have collected the data. In our case, we received a national database, validated by national services. Our expertise on how health services work allows us to know accurately how the information was collected. However, the exact details regarding the collection of information for this particular database and its association with the lack of information at each step was not provided by national services, despite our efforts and communication attempts with the Health Directorate services that provided the database. Therefore, for each variable, the category “unknown” corresponds to a lack of information. Regarding preconditions, the category “no-present” includes patients without that precondition registered in the database, either because they do not suffer from that precondition, or because no information was registered. This limitation was mentioned among the data limitations concerning the database in the Discussion section.
Changes: Discussion section, line 160-164: “A further limitation is related to the data that concerns the initial phase of the pandemic in Portugal up to April 21. The pandemic is still ongoing, registering on June 25 about 40866 infected individuals and 1555 deaths [43]. Nevertheless, no update of the data was made available up to the moment and therefore the hypothesis of a difference in mortality between the initial and actual phase remains open.”
Minor revisions:
ABSTRACT
From my point of view, the abstract could be improved:
3- Briefly explain why the identification of risk factors of death is important.
Response: The authors thank the Reviewer’s indication. We included the information as requested.
Changes: Abstract section, line 23-25: “The identification of risk factors is important to allow more efficient public health and health services strategic interventions with significant impact on deaths by COVID-19.”
4- Clearly indicate when the primary outcome was measured.
Response: The authors thank the Reviewer’s query. As indicated in the response to points 1 and 2, the primary outcome was measured (registered and confirmed) by Medical Doctors serving as health delegates on the daily death registry. The information was not available in the dataset nor the authors received the information from the Health Directorate despite several contacts. Therefore, it was assumed that the primary outcome measure was measured in the period between January 1 and April 9, 2020 as indicated in the manuscript.
Changes: Abstract section, line 27: “A national dataset with the first 20293 patients infected with COVID-19 between January 1 and April 9, 2020 was analyzed. “
Line 28: “The primary outcome was measured (registered and confirmed) by Medical Doctors serving as health delegates on the daily death registry. “
5- "with binomial error distribution and logit link function", "The significant level was set at 5%": These methods elements are probably not essential in the abstract.
Response: The authors thank the Reviewer’s suggestion. The methods elements were removed from the Abstract as indicated.
Changes: Abstract section, lines 30, 33, and 33: “with binomial error distribution and logit link function", "The significant level was set at 5%", both deleted.
6- Rather than indicate the AUC of the model in the abstract, I would suggest the authors indicate the OR for identified risk factors. Indeed, this study aimed to identify risk factors for death, not to create a prognostic model.
Response: The authors thank the Reviewer’s suggestion. The methods elements were removed from the Abstract as indicated.
Changes: Abstract section, lines 31-32 “The model performance was illustrated using the area under the curve (AUC).” deleted.
Lines 34-39: “sex (male: OR = 1.47)), age ( (55—60] years, OR = 6.01; (60—65] years, OR = 10.5; (65—70] years, OR = 20.4; (70—75] years, OR = 34; (75—80] years, OR = 50.9; (80—85] years, OR = 70.7; (85—90] years, OR = 83.2; (90—95] years, OR = 91.8; (95—104] years, OR = 140.2), Cardiac disease (OR = 2.86), kidney disorder (OR = 2.95) and Neuromuscular disorder (OR = 1.58); while condition (None [absence of precondition]; OR = 0.49)”
RESULTS
The tables are difficult to read in the present form.
7- Please supress all the unneccessary points in the tables.
Response: The authors thank the Reviewer’s suggestion. The problem with the tables was related to a conversion error. The authors amended the tables as requested.
Changes: Line 177, Table 1; Line 193, Table 2; Line 215, Table 3.
8-In table 1, please clarify or suppress the part "outcome" of the table: What do you mean with "ongoing treatment" for 91.3% of patients? It is unlikely that patients infected in January are still under specific treatment, though this depends on when the data are extracted.
Response: The authors thank the Reviewer’s query. The “ongoing treatment” is conditional to April 9, 2020, where at that point there were patients that were either recovered, died, were in ongoing treatment or of unknown status. The authors indicated in Table 1 title that this information is conditional to the date of April 9, 2020.
Changes: Lines 177 and 178, Table 1
9- In table 3, please supress the full lines between the different age classes.
Response: Thank you. Proof read and corrected.
Changes: Results section, Line 215, Table 3
10-Figure 1, please explicit the abbreviations, or better change the name of the variables.
Response: The authors thank the Reviewer’s suggestion. We removed the abbreviations and capital letters of the variables.
Changes: Results section, Line 1, Figure 1.,
DISCUSSION:
11-Line 22: age (continuous..age>75 ans) + Line 23 "age (18-55)" > I would simplify by indicating that advanced age is a risk factor.
Response: The authors thank the Reviewer’s indication. The text was adapted as suggested.
Changes: Discussion section, line 28: introduction of “advanced” deletion of “(continuous, age65-75; age >75 years”.
12-Line 50: Wether age is per se a risk factor for death in pneumonia after adjustment for comorbidities remain a debated question. I would suggest further discussing it in light of the current literature.
Response: The authors thank the Reviewer’s indication. The authors further extended the Discussion on this topic as suggested by the Reviewer.
Changes: Discussion section, Lines 57-73: “Nevertheless, age remained a risk factor for death in our study even in patients without preconditions as it would be expected, given that advanced age is a risk factor for death even in the absence of COVID-19 infection. Indeed, it is possible that the increase in risk of death is more due to ageing than to COVID-19 infection proper. Still, the link between older individuals and the likelihood to develop severe and critical cases of COVID-19 has been made before, either due to immunosenescence, malnutrition, or ignoring more easily the early symptoms and consequently missing the best time to seek medical advice [20,21]. Furthermore, several studies registered an important association between increased age and COVID-19 severity/fatality [22-26]. The vulnerability of the elderly individuals is illustrated in recent studies. A retrospective observational study investigating mortality in hospitalized patients with COVID-19 registered the great vulnerability of patients residing in retirement homes, with older age independently associated with mortality when adjusted for other variables of interest [22]. On the opposite residence condition, the shape of COVID-19 vulnerability was estimated based on a random infection of 10% of the population living in private households (excluding individuals living in retirement homes) of 81 countries [27]. In this study it was estimated that national age and coresidence patterns can alter the vulnerability of a country to COVID-19 outbreaks, with direct effects dependent on a country’s age structure and indirect effects dependent of the size and age structure of a country’s households [27].”
13-Line 81: The authors clearly highlight that preexisiting cardiovascular disease remained the main risk factor for death. Please further discuss the link between pneumonia and cardiovascular complications, including type 2 myocardial infarction, particularly frequent in this older population.
Response: The authors thank the Reviewer’s indication. The authors further extended the Discussion on this topic as suggested by the Reviewer.
Changes: Discussion section, lines 108-113: “Furthermore, the link between pneumonia and cardiovascular complications should be accounted for: Recent studies explore/registered myocardial injury during SARS-CoV-2, secondary to type 2 myocardial infarction, consequence of increased oxygen demand or reduced oxygen supply during respiratory failure [34-36]. In this scenario, cytokines microvascular activation can cause not only myocardial injury but also harm other systems involved in COVID-19 infections including the kidneys [37].”
CONCLUSIONS:
14-I am not sure the distinction between age over 75y and age 18-55 is relevant.
Response: The authors thank the Reviewer’s suggestion. The information was deleted from the Conclusions section as requested and adapted in the Discussion section
Changes: Discussion section, lines 45-48: deletion of “both in continuous and in subgroup analysis, where individuals of less than” and “had 3.33 times lesser chance of dying, while individuals between 65 and 75 years and over 75 years of age had a 96% and 3-fold increase in lethality on the adjusted model, respectively”; insertion of “with lethality adjusted ORs largely increasing after”
Conclusions section, line 173, deletion of “over 75 years” and insertion of “advanced”; lines 174 and 175, deletion of “; while age [18,55) years was associated with a decreased risk”
15-Please indicate Author Contributions and Funding sources.
Response: The authors thank the Reviewer’s indication. The author contributions were introduced. The study received no funding. The information was introduced as requested.
Changes: Lines 181 to 185, Author Contributions: “Conceptualization, P.J.N.; Data curation, P.J.N.; Formal analysis, P.J.N.; Investigation, P.J.N.; M.A.N. A.S.C.; R.M.R.; L.B.N.; C.F.; C.C.; M.L.; R.A.; A.V.C.; Methodology, P.J.N.; M.A.N. A.S.C.; Supervision, P.J.N.; R.M.R; A.V.C.; Validation, P.J.N.; A.V.C.; Visualization, P.J.N.; M.A.N. A.S.C.; R.M.R.; L.B.N.; C.F.; C.C.; M.L.; R.A.; A.V.C.; Writing—original draft, P.J.N.; M.A.N. A.S.C.; R.M.R.; L.B.N.; C.F.; C.C.; M.L.; R.A.; Writing—review & editing, P.J.N.; M.A.N. A.S.C.; R.M.R.; L.B.N.; C.F.; C.C.; M.L.; R.A.; A.V.C.”
Line 186, Funding: “The study was not funded by any public or private institution”
Reviewer 2 Report
Dear authors,
many thanks for this interesting article.
I would like to raise a few points here.
The link to the dataset which is publicaly available should be included, also for cross checking and other researchers.
It would be most useful to have a focus on a multivariant analysis approach.
The end date is 9. April. How is the current situation in Portugal. Maybe it could be beneficial to include a broader timeframe. It may also be speculated that in the initial phase the mortality was different than in a later one or vice versa.
The data has also some geographical data included: the reader should be aware what the make up of the regions is, e.g. populations, cases/population.
One other point is that the Tables should be revised.
Figure 1 Asthma, I would also suggest that the different factors should not be in capital letters.
Author Response
Response to Review of manuscript no. jcm-849655, “The role of health preconditions on COVID-19 deaths in Portugal: Evidence from surveillance data of the first 20293 infection cases”
Reviewer 2
Dear authors,
many thanks for this interesting article.
I would like to raise a few points here.
1- The link to the dataset which is publicaly available should be included, also for cross checking and other researchers.
Response: The authors thank the Reviewer’s indication. The dataset is not publicly available to all the public, as the Health Directorate in Portugal only allowed access to research teams that provided an ethical committee authorization to perform research on the dataset. The authors introduced the link used to request access to data
Changes: Introduction section, line 89-91: “The dataset is not available to the public, as the Health Directorate in Portugal only allowed access to research teams that provided an ethical committee authorization to perform research on the dataset [10].”
2-It would be most useful to have a focus on a multivariant analysis approach.
Response: The authors thank the Reviewer’s suggestion. A multivariate statistical inferential approach was the basis of our study: logistic regression models were applied using a generalized linear model with binomial error distribution and logit link function to estimate the crude and adjusted Odds Ratio (OR) with 95% confidence intervals (95%CI) for each potential risk factor. Table 3 presents the univariate logistic models results on the two left columns and the multivariate models results on the following 6 columns. The results on several multivariate logistic models are presented: including only individuals without preconditions, with variables sex and age; using preconditions adjusted for Age and Sex; the full model also adjusted for Pregnancy. The authors introduced this information in the text to make it clearer. Furthermore, the Abstract was adpated to focus on multivariable analysis approach, removing information about the AUC and introducing the OR’s.
Changes: Abstract section, lines 31-32 “The model performance was illustrated using the area under the curve (AUC).” deleted.
Lines 34-39: “sex (male: OR = 1.47), age ( (55—60] years, OR = 6.01; (60—65] years, OR = 10.5; (65—70] years, OR = 20.4; (70—75] years, OR = 34; (75—80] years, OR = 50.9; (80—85] years, OR = 70.7; (85—90] years, OR = 83.2; (90—95] years, OR = 91.8; (95—104] years, OR = 140.2), Cardiac disease (OR = 2.86), kidney disorder (OR = 2.95) and Neuromuscular disorder (OR = 1.58); while condition (None [absence of precondition]; OR = 0.49)”
Results section, lines 197-199: “Table 3 registers three multivariable models: A model only including patients without preconditions, a model with preconditions adjusted for age and sex, and the full model.”
3-The end date is 9. April. How is the current situation in Portugal. Maybe it could be beneficial to include a broader timeframe. It may also be speculated that in the initial phase the mortality was different than in a later one or vice versa.
Response: The authors thank the Reviewer’s query. The data concerns the initial period of the pandemic evolution in Portugal. The situation is still ongoing, registering on June 25 about 40866 infected individuals and 1555 deaths. No update of the data was made available up to the moment. The authors performed complementary calculations to address this issue, introducing two multivariable models: one model for the cases up to March 31, 2020 (excluding cases from the last 3 weeks) where the effect of age on death by COVID-19 increased; and another model for Hospitalized cases, where both age and precondition effects decreased. The general results of these models point to a reasonable consistency of the full multivariable model irrespective of the dates limitation. The authors introduced the information on the Results section and on the Discussion section on the study limitations.
Changes: Results section, line 209-214: “Complementary calculations (multivariable models) were performed with the objective of evaluating the data robustness and model consistency (Table 4). First, we analyzed a model excluding data from the last 3 weeks, so that cases with insufficient follow-up were excluded. Next, we analyzed a model including only the Hospitalized cases, assuming that these cases would have more accurate recording of comorbidities. As shown in Table 4, the results of these two reduced models are consistent with the model for the full dataset in Table 3.”
Results section, line 6: Table 4.
Discussion section, lines 157-164: “The authors performed complementary calculations to evaluate data robustness and model consistency: The results of these complementary estimations (one model excluding cases from the last 3 weeks and the other model on hospitalized cases) show reasonable consistency with the full multivariable model. A further limitation is related to the data that concerns only the initial phase of the pandemic in Portugal up to April 21. The pandemic is still ongoing, registering on June 25 about 40866 infected individuals and 1555 deaths [43]. Nevertheless, no update of the data was made available up to the moment and therefore the hypothesis of a difference in mortality between the initial and actual phase remains open.”
4-The data has also some geographical data included: the reader should be aware what the make up of the regions is, e.g. populations, cases/population.
Response: The authors thank the Reviewer’s suggestion. The geographical information concerning the populations, cases/population for each region were introduced as suggested.
Changes: Results section, lines 149-157: “The distribution of infection rates per 100,000 inhabitants was the following: 341.8 in the North region (12211 individuals infected in a population of 3572583); 107.6 in the Algarve region (472 individuals infected in a population of 438864); 127.1 in the Center region (2817 individuals infected in a population of 2216569); 149.8 in the Lisbon Metropolitan region (4264 individuals infected in a population of 2846332); 55.4 in the Alentejo region (391 individuals infected in a population of 705478); 18.9 in the Madeira region (48 individuals infected in a population of 253945); and 37.1 in the Azores region (90 individuals infected in a population of 242846).
5-One other point is that the Tables should be revised.
Response: The authors thank the Reviewer’s suggestion. The problem with the tables was related to a conversion error. The authors amended the tables as requested.
Changes: Line 177, Table 1; Line 193, Table 2; Line 215, Table 3.
6-Figure 1 Asthma, I would also suggest that the different factors should not be in capital letters.
Response: The authors thank the Reviewer’s suggestion. We removed the abbreviations and capital letters of the variables.
Changes: Results section, Figure 1, line 1.
Round 2
Reviewer 2 Report
Dear authors,
all my points were addressed.
With best regards